# Polygenic Panels Predicting the Susceptibility of Multiple Upper Aerodigestive Tract Cancer in Oral Cancer Patients

**DOI:** 10.3390/jpm11050425

**Published:** 2021-05-18

**Authors:** Huei-Tzu Chien, Chi-Chin Yeh, Chi-Kuang Young, Tzu-Ping Chen, Chun-Ta Liao, Hung-Ming Wang, Kai-Lun Cho, Shiang-Fu Huang

**Affiliations:** 1Department of Nutrition and Health Sciences, Chang Gung University of Science and Technology, Tao-Yuan 33302, Taiwan; kathy.htchien@gmail.com; 2Research Center for Chinese Herbal Medicine, College of Human Ecology, Chang Gung University of Science and Technology, Tao-Yuan 33302, Taiwan; 3Master Program in Applied Molecular Epidemiology, College of Public Health, Taipei Medical University, Taipei 11031, Taiwan; ccyeh@tmu.edu.tw; 4Department of Public Health, College of Public Health, China Medical University, Taichung 40402, Taiwan; 5Cancer Center, Wan Fang Hospital, Taipei Medical University, Taipei 11696, Taiwan; 6Department of Otolaryngology, Chang Gung Memorial Hospital, Keelung 20401, Taiwan; rioriorioman@gmail.com; 7Department of Thoracic Surgery, Chang Gung Memorial Hospital, Keelung 20401, Taiwan; kkl3490@yahoo.com.tw; 8Department of Otolaryngology, Head and Neck Surgery, Chang Gung Memorial Hospital, Linkou 33342, Taiwan; liaoct@adm.cgmh.org.tw (C.-T.L.); kailuncho@gmail.com (K.-L.C.); 9Medical College, Chang Gung University, Tao-Yuan 33302, Taiwan; whm526@adm.cgmh.org.tw; 10Division of Hematology/Oncology, Department of Internal Medicine, Chang Gung Memorial Hospital, Tao-Yuan 33342, Taiwan; 11Graduate Institute of Clinical Medical Science, Chang Gung University, Tao-Yuan 33302, Taiwan

**Keywords:** SNP array, single nucleotide polymorphism, head and neck cancer, multiple primary cancer, upper aerodigestive tract

## Abstract

Head and neck cancer was closely related with habitual use of cigarette and alcohol. Those cancer patients are susceptible to develop multiple primary tumors (MPTs). In this study, we utilized the single nucleotide polymorphisms (SNPs) array (Affymetrix Axion Genome-Wide TWB 2.0 Array Plate) to investigate patients’ risks of developing multiple primary cancers. We recruited 712 male head and neck cancer patients between Mar 1996 and Feb 2017. Two hundred and eighty-six patients (40.2%) had MPTs and 426 (59.8%) had single cancer. Four hundred and twelve normal controls were also recruited. A list of seventeen factors was extracted and ten factors were demonstrated to increase the risks of multiple primary cancers (alcohol drinking, rs118169127, rs149089400, rs76367287, rs61401220, rs141057871, rs7129229, older age, rs3760265, rs9554264; all were *p* value < 0.05). Polygenic scoring model was built and the area under curve to predict the risk developing MPTs is 0.906. Alcohol drinking, among the seventeen factors, was the most important risk factor to develop MPT in upper aerodigestive tract (OR: 7.071, 95% C.I.: 2.134–23.434). For those with high score in polygenic model, routine screening of upper digestive tract including laryngoscope and esophagoscope is suggested to detect new primaries early.

## 1. Introduction

In Taiwan, due to the habitual use of cigarette, alcohol and areca-quid (AQ), oral cavity cancer was the fourth most common cancer in males. The annual increased rate of oral cancer was more than 2%, with an increased rate of 26.2% from 2007 to 2011 [1]. It was one of the most rapid growing malignancies in Taiwan [1]. The primary treatment for oral cavity squamous cell carcinoma (OSCC) is radical surgery with or without post-operative chemoradiation [2]. For individuals’ genetic susceptibility of OSCCs, there were studies focusing on the polymorphisms in enzymes carcinogen metabolism [3,4] and/or in the capacity of DNA repair [5,6,7]. For example, genetic polymorphisms in XPD, XRCC1, and XRCC3 genes increased the risks of OSCCs. ERCC2 (XPD) Gln/Gln homozygote also increased susceptibility of the Asian population in digestive tract cancers [8]. Similarly, cytochrome P450 (*CYP*) enzymes are monooxygenases that catalyze many reactions involving carcinogens [9,10]. Pharyngeal cancer patients were reported to have a higher frequency of the CYP1A1 Val/Val genotype than controls [11].

As the disease becomes stable after treatment, the risk of developing a new primary tumor becomes higher. Head and neck cancer patients are vulnerable to develop multiple primary tumors (MPTs) according to the theory of “field cancerization” [12]. The chance to develop a new primary in upper aero-digestive tract is about 15% within 5 years after first primary cancer treatment [13].

The frequency of MPTs was 70.3% at the oral cavity, and the others occur in the oropharynx, hypopharynx or esophagus. Most of the MPTs are located in the upper aero-digestive tract. Patients living with habitual use of cigarettes, alcohol and AQ had a 123-fold increased risk of developing OSCC [14]. The mucosa, after exposure to environmental carcinogens, is vulnerable to developing pre-malignant alteration. It includes submucosal fibrosis, leukoplakia and erythroplakia in the aero-digestive tract. These render the OSCC patients vulnerable to developing multiple oral cancers. We found that patients having habitual of AQ use had higher chance of MPT development [13]. In a recent study, a lower survival rate was demonstrated in patients with local recurrences with respect to patients with MPTs [15]. In Taiwan, due to the use of AQ in OSCC patients, the frequency of developing MPTs is higher than in other areas [13].

Previously, we identified that alcohol metabolizing enzymes play roles in the tumor formation in upper aerodigestive tract (ADT) cancers in head and neck cancer patients [16]. In this study, we intended to search novel genetic polymorphism loci predicting the susceptibility of MPTs in OSCC patients on a genome-wide scale.

## 2. Materials and Methods

### 2.1. Study Population

All cancer cases were histologically confirmed as squamous cell carcinoma in oral the cavity and patients signed informed consent for participation before surgery. Normal controls were recruited from participants in a serum lead level study in healthy workers [16]. Participants for this study were recruited from all patients who received treatment from the Head and Neck Surgery Department at Chang Gung Memorial Hospital, Lin-Kuo (Tao-Yuan, Taiwan) between 1996–2018. Information about patients’ habitual exposure history, including cigarette smoking, alcohol drinking and AQ chewing, as well as general demographic information, were obtained by uniform interview conducted by a well-trained technician using a questionnaire. Ever smokers were defined as those who have smoked for more than 100 cigarettes in their lifetime. Ever alcohol drinkers were defined as those who drank at least once per month. Habitual AQ chewing was defined as those who have chewed more than 100 nuts in their lifetime. This study was approved by the Institutional Review Board of Chang Gung Memorial Hospital and was undertaken according to the ethical guidelines of human investigation.

All the oral cancer patients were diagnosed by pathology proof. The tumor subsites were listed in Table 1. At the time of primary tumor diagnosis, all the patients received pre-treatment workup. They include whole body positron emission tomography and/or panendoscopy examination to evaluate the occurrence of synchronous upper ADT cancers. After the treatment of index cancer, the patients received regular follow-up in the clinic. If the patient was found to have new primary lesions in the oral cavity, they would be categorized as multiple oral cavity cancer. If the tumor lesions were in oropharynx, hypopharynx or esophagus, they would be categorized as multiple upper ADT cancers. If the patients did not develop new primaries during follow-up (ranged from 0 to 252.0 months, mean 52.7 months), they would be categorized as single oral cavity cancer. All the patients with new lesions were first surveyed to exclude the possibility of tumor recurrence or tumor metastasis.

### 2.2. DNA Extraction

For each participant, 10 mL of venous blood was drawn; separated into plasma, buffy coat cells and red blood cells by centrifugation within 18 h of obtaining the blood, and stored separately at –80 °C. From the buffy coat cell, genetic DNA was extracted for genotyping. DNA extraction was done using the standard phenol-chloroform method. To evaluate the quantity of DNA and assess purity index, the NanoDrop ND-1000 spectrophotometry (NanoDrop Technologies, Wilmington, DE, USA) was used and a ratio absorbance of 260/280 and a purity index of >1.8 was considered optimal. The volume for array analysis was 50 μL and concentration will be 15 ng/μL for each sample.

### 2.3. SNP Array Genotyping and Quality Control

All samples for array analysis were processed on Axiom Genome-Wide TWB Array Plate by Affymetrix GeneTitan using the GeneTitan automatic instrument (Thermo Fisher Scientific, MA, USA). Genotype calling was performed using the standard procedure with the default parameters as recommended by platform manufacturer. Genotyping was performed at the National Center for Genome Medicine (NCGM), Academia Sinica, Taipei, Taiwan.

Quality control was performed at sample and marker level. The control procedure was performed first with each individual, including sample quality, kinship, and population stratification. Dish sample quality control (DQC) was used to monitor non-polymorphic location to specify signal and background channels. Subjects whose DQC values are unsatisfactory were discarded and those with less than 97% call rate were excluded. To check for plate-wise genotyping biases, plate pass rate was calculated as samples passing DQC and 97% call rate divided by total samples on the plate. All samples with a call pass rate >95% and an average call rate of sample passing >99% were included in the analysis. Inbreeding coefficients were assessed and samples found with string kingship were excluded. We used multidimensional scaling analysis on the genome-wide identity and any outliers away from the clustering were eliminated. At the marker level, markers failing the following thresholds: missing rate (<2%), minor allele frequency rate (>5%) or Hardy–Weinberg Equilibrium (*p* > 0.0001) were excluded. The replication sample follows the same quality control.

### 2.4. Statistics for Array Analysis

A genome-wide association scan on Taiwanese ancestry was performed by using Affymetrix Axion Genome-Wide TWB 2.0 Array Plate. Nucleotide polymorphism markers for MPTs were selected by logistic regression under the additive genetic model. Susceptible polymorphism gene panels for MPTs were generated by Least Absolute Shrinkage and Selection Operator (LASSO) penalized regression model (SPSS version 25.0, IBM, New York, NY, USA) analysis and incorporating gene-environment interaction. The risks for SNP genotypes to develop MPTs were calculated by using logistic regression analyses with PLINK software (v1.90) [17,18] (Table 2 and Table 3).

### 2.5. Polygenic Risk Score

The Polygenic Risk Score (PRS) was calculated for all included individuals using the formula:PRS = β_1_ X_1_ + β_2_ X_2_ + … + β_n_ X_n_
where β is the per-allele log OR for MPTs associated with the risk allele of the SNP, X is the number of risk alleles per SNP (0, 1, or 2) and *n* is the total number of SNPs included in the PRS [19].

We plotted Receiver-Operating Characteristic curves to compare the score of susceptible factors to predict the development of multiple primary cancers. Cutoff values for the score were obtained from ROC-curves and Youden index (Youden index = sensitivity + specificity − 1) [20].

## 3. Results

### 3.1. Characteristics of Study Population

We recruited 712 patients with primary cancers. The subsite distribution of the patients are listed in Table 1. In the further analysis, we excluded 5 female patients from further analysis to minimize the effect of gender. All the normal controls were male patients without cancer at the time of blood sampling. In head and neck cancer patients, 426 (59.8%) had single primary cancer during follow-up (ranged from 0 to 252.0 months, mean 52.7 months). Two hundred and eighty-six patients developed MPTs in which 111 (38.8%) were located only in oral cavity, 120 (42.0%) in upper ADT and 55 (19.2%) in other sites other than oral and upper ADT.

### 3.2. Upper Aerodigestive Tract MPT Related SNPs

In the first step of analysis, we compared the genotype differences by χ^2^ between cancer patients and normal patients. A total of 27,532 SNPs generated from the analysis. The SNPs were posed as a gene set to compare between patients with upper aerodigestive tract MPT and with single oral cavity cancer. PLINK software was used to extract the gene set for comparison. A total of 1563 SNP loci were generated. We set *p* < 0.001 and odds ratio (ORs) ≥2 or ≤0.5 as the cut-off value to select upper aerodigestive MPT related genes. Also, the minor allele frequency (MAF) that is less than 1% were excluded from further analysis. After the filtering of second step, a total of 45 SNP loci were generated (Appendix A).

### 3.3. Multivariate Analysis

The 45 genes were further analyzed by LASSO algorithm to reduce the variables and choose the most related with upper ADT-MPTs. We also put cigarette smoking, AQ chewing, alcohol drinking and age (<50 years old vs. ≥50 years old) into analysis. The variables were reduced to 20 variables. The frequencies of eighteen SNP loci between patients with upper ADT-MPTs were found to be significantly different from patients with oral cavity cancer patients alone (Table 2). For the next step, we adopted logistic regression to calculate the ORs of the factors and SNP loci to develop multiple ADT cancers. The results are shown in Figure 1 and Table 3 by using the forest plot. All the SNPs and factors were *p* value ≤ 0.05.

### 3.4. Polygenic Risk Score Construction

Each gene and factor were given one score according to the polygenic risk score in the materials and methods. The score (β) is the per-allele log OR to develop MPTs. The summation of the score was correlated with the development of multiple ADT cancers by using ROC curves (Figure 2). We calculated the sensitivity, specificity and Youden index to select the best cut-off point for this model (Table 4). The best cut-off point is the 2.426500 and the AUC is 0.906.

## 4. Discussion

Genome-wide scans represent an opportunity to identify common genetic variants that predispose to human disease. Recently, GWAS have emerged as a powerful agnostic approach for identifying novel susceptibility loci for many forms of cancers. Genetic variants have been investigated in many OSCC association studies based on either candidate or multiple genes, with inconclusive results. Large scale studies on loci associated with OSCC are rare. The study to investigate the group of MPT is difficult. The reason is due to the difficulty of recruiting and collecting the OSCC patients developing MPTs. 

The majority of MPTs occurred years after the treatment of primary index tumor were categorized as meta-chronous tumor [21,22]. The SNP loci filtered to be correlated with MPT in Table 3 were mostly form the metachronous MPTs. In our patients, 33 patients had synchronous MPTs in the upper aerodigestive tract. These patients were diagnosed of multiple cancers in oral cavity, oropharynx or esophagus within 6 months of the index cancer [23,24]. The most significant correlated SNPs were rs3760265 (*p* = 0.008, HR = 2.714, 95% C.I. = 1.275–5.779), rs848 (*p* = 0.017, H.R. = 2.653, 95% C.I. = 1.159–6.076) and rs9554264 (*p* = 0.026, H.R. = 2.250, 95% C.I. = 1.085–4.666).

Most of the genes selected in our study were SNP loci from intron in genes. Some SNPs from our polygenic panel could be identified with known genes from the upstream introns (Table 5). Three of the SNP loci were reported to have functions. rs7129229 was an intron variant FAT3 gene. In human cancer, a point mutation in Fat3 [25] results in pancreatic tumors. The human surfactant protein C promoter helps to regulate the expression of Fat3, and its expression is downregulated in non-small cell lung cancer. The frequency of copy number alteration or mutation of FAT3 were also increased in the Indian ICGC (International Cancer Genome Consortium) cohort [26]. rs7847271 is another intron variant TNCT. This gene encodes an extracellular matrix protein with a spatially and temporally restricted tissue distribution. It was also reported to be related with prostate cancer [27]. rs3760265 was an intron variant in CACNG5 (Calcium Voltage-Gated Channel Auxiliary Subunit Gamma 5). The protein encoded by this gene is a type II transmembrane AMPA receptor regulatory protein (TARP). TARPs regulate both trafficking and channel gating of the AMPA receptors. This gene is a susceptibility locus for schizophrenia and bipolar disorder [28]. Most of the genes of SNPs were not yet related to head and neck cancers. Future studies are needed to validate the efficacy of our polygenic models for MPTs in upper aerodigestive tract.

Alcohol drinking played a major role in the formation of head and neck cancer. The risk increased more evident in oropharynx and hypopharynx cancer as compared with the oral cavity [29]. Alcohol, as we reported in our previous study, also plays an important role in the development of multiple ADT cancer in Taiwan [16]. In our study, it is still the most important risk factor for MPT in upper aerodigestive tract (Table 3, OR = 7.071, 95% CI: 2.134–23.434). In Asians, the role of alcohol drinking is more evident in oropharynx and esophageal cancer. This is probably related to ethnic difference in alcohol metabolism of the genes ALDH2 and ADH1B [30]. Aged patients (>50 years old) was another parameter that predicts high risks to develop MPT. The middle aged OSCC patients were reported to have a better prognosis [31,32,33]. It is thus easier to carry a higher risk of multiple upper aerodigestive tract cancers in this group of patients.

The limitation of this study is that all the multiple ADT cancer developed a few years after treatment. The detection and diagnosis of ADT malignancy depends on patients’ survival and regular follow-up. It could under-estimate the real incidence of multiple ADT malignancies. However, this is the largest cohort that recruited the largest number of oral cavity cancer patients and most thorough follow-up with examination to date. As per retrospective in this study, a validation cohort especially prospective in the recruitment of patients is needed.

Routine screening of upper ADT is a method to detect early multiple cancer. From our study, we can calculate the susceptible score by polygenetic panels of OSCC patients at the time of diagnosis. For high-risk patients, regular fiberoptic examination or esophagus screening is suggested in them. Those with low susceptible score can be followed up by CT/MRI after surgery. The frequency of testing can be based on institutional guidelines. Or, upper ADT panendoscope examination can be arranged when patient had dysphagic symptoms. Selected high risks patients should have more frequent clinic visit and screening.

The identification of SNPs in key genes involved in carcinogenesis has been a major area of study that can provide predictive biomarkers for precision preventive medicine in oral cancer. Patients who smoke, chew AQ or drink alcohol had different alterations of mucosa. It can be used as a marker to predict susceptibility of an individual to develop oral cavity malignancy or MPTs. Such knowledge could help determine more precise OSCC screening, resulting in earlier detection and prompt treatment, better response to treatment and eventually better survival.

## 5. Conclusions

We successfully screened a polygenic panel of high-risk genes for developing multiple primary cancers in upper ADT, in which alcohol drinking played an important role. For those with high risk allele carriers, routine screening of upper ADT—including laryngoscope and esophagoscope—is suggested to detect new primaries early.

## Figures and Tables

**Figure 1 jpm-11-00425-f001:**
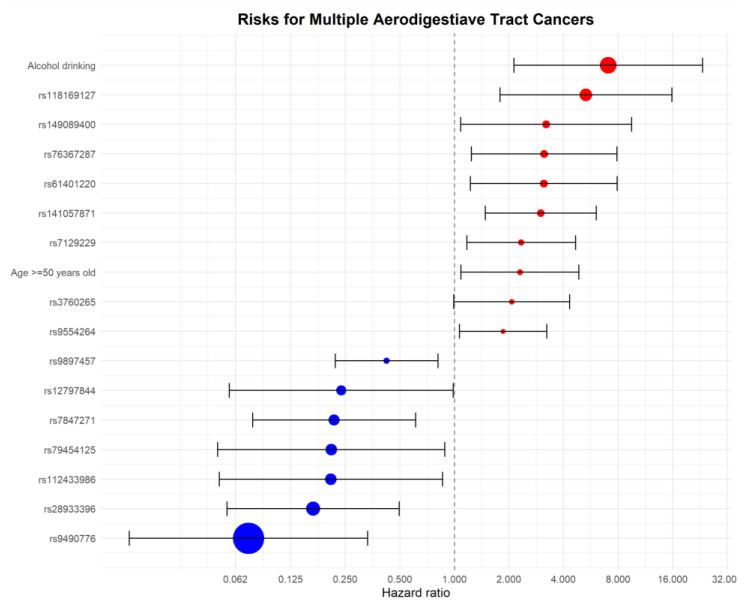
Hazard ratio to develop multiple aerodigestive tract cancer in oral cancer patients. (Red: increased risk; blue dot: decrease risk). The hazard ratio is represented by the size of the dot.

**Figure 2 jpm-11-00425-f002:**
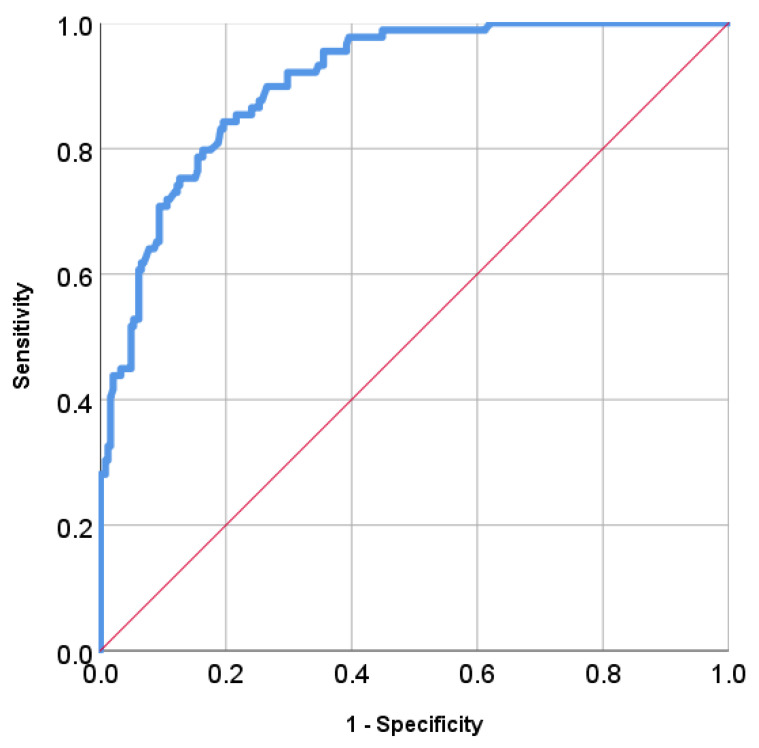
ROC curves calculating the best-fit number of genes in predicting multiple aerodigestive tract cancer in oral cancer patients.

**Table 1 jpm-11-00425-t001:** The demographic data in the cohort of primary and multiple primary tumors for SNP array study (*n* = 712).

Characteristics	Cancer Patients [*n* (%)]	Normal Patients [*n* (%)]	*p* Value
Age (±S.D.)	52.58 (±10.92)	43.66 (±15.67)	
Range	25–88	18–99	
Tumor subsites			
Oral cavity cancer	364 (51.1)		
Oropharynx	135 (19.0)		
Hypopharynx	190 (29.7)		
Larynx	14 (0.2)		
Nasal cavity	2 (0.3)		
Skin	1 (0.1)		
Esophagus	6 (0.8)		
Gender			
Male	707 (99.3)	398 (100.0)	0.168 *
Female ^1^	5 (0.7)	0 (0)	
Alcohol drinking			
Yes	481(79.9)	264 (66.3)	<0.001
No	121(20.1)	134 (33.7)	
AQ chewing			
Yes	498 (82.6)	82 (20.6)	<0.001
No	105 (17.4)	316 (79.4)	
Cigarette smoking			
Yes	549 (91.0)	228 (57.3)	<0.001
No	54 (9.0)	170 (42.7)	
Number of malignances			
Single primary	426 (59.8)		
Multiple primary tumors	286 (40.2)		
Multiple oral cavity cancers	111 (38.8)		
Multiple upper aerodigestive tract MPTs	120 (42.0)		
Non-oral and non-aerodigestive tract MPTs	55 (19.2)		

^1^ Female patients were excluded in analysis. *: Fisher’s exact test.

**Table 2 jpm-11-00425-t002:** The frequency of susceptible SNP locus in patients.

CHR	SNP	Minor Allele	Frequency (ADT)	Frequency (Oral Cancer)	Major Allele	Frequency (Normal Population)
1	rs118169127	C	0.088	0.02181	T	0.06675
2	rs112433986	-	0.024	0.0755	A	0.08629
5	rs848	A	0.4355	0.3138	C	0.2834
6	rs79454125	C	0.024	0.07191	T	0.04672
6	rs9490776	A	0.02846	0.09532	G	0.1083
7	rs76367287	C	0.124	0.04849	A	0.09068
9	rs141057871	A	0.2195	0.1221	G	0.1322
9	rs7847271	A	0.04	0.1271	G	0.07305
10	rs11255400	A	0.028	0.08361	G	0.03526
11	rs12797844	C	0.02016	0.06376	T	0.02519
11	rs7129229	T	0.2358	0.1421	A	0.1344
13	rs9554264	T	0.348	0.2258	C	0.306
14	rs1959792	T	0.2621	0.151	C	0.1679
16	rs61401220	-	0.108	0.04348	T	0.03914
17	rs9897457	C	0.148	0.2559	T	0.1856
17	rs3760265	T	0.188	0.09866	C	0.154
19	Affx-15929578 (rs28933396)	A	0.03361	0.1747	G	0
23	rs149089400	A	0.08	0.01974	G	0.01741

**Table 3 jpm-11-00425-t003:** Logistic regression of the candidate genes in predicting the risks for formation of multiple primary cancers.

Name	β	*p* Value	Hazard Ratio (95% Confidence Interval)
Alcohol drinking	1.956	0.001	7.071 (2.134–23.434)
rs118169127	1.671	0.003	5.320 (1.785–15.854)
rs149089400	1.165	0.036	3.206 (1.082–9.500)
rs76367287	1.139	0.016	3.123 (1.239–7.874)
rs61401220	1.137	0.017	3.117 (1.227–7.921)
rs141057871	1.098	0.002	2.998 (1.482–6.062)
rs7129229	0.849	0.016	2.336 (1.170–4.665)
Age ≥ 50 years old	0.834	0.029	2.302 (1.088–4.868)
rs3760265	0.728	0.053	2.071 (0.990–4.332)
rs9554264	0.618	0.029	1.856 (1.066–3.230)
rs9897457	−0.862	0.010	0.422 (0.220–0.812)
rs12797844	−1.437	0.048	0.238 (0.057–0.985)
rs7847271	−1.529	0.004	0.217 (0.077–0.610)
rs79454125	−1.566	0.034	0.209 (0.049–0.886)
rs112433986	−1.570	0.030	0.208 (0.050–0.858)
rs28933396	−1.795	0.001	0.166 (0.056–0.496)
rs9490776	−2.617	0.001	0.073 (0.016–0.332)

β: log—hazard ratio.

**Table 4 jpm-11-00425-t004:** The best number of factors predicting the development of multiple aerodigestive tract cancers.

Positive if Greater than or Equal to	Sensitivity	1—Specificity	Specificity	Youden Index (Sensitivity + Specificity − 1)
−7.655000	1.000	1.000	0.000	0.000
−6.304500	1.000	0.996	0.004	0.004
−5.194500	1.000	0.988	0.012	0.012
−4.094000	1.000	0.967	0.033	0.033
−3.065500	1.000	0.902	0.098	0.098
−2.113000	1.000	0.853	0.147	0.147
−1.190000	1.000	0.763	0.237	0.237
−0.100500	1.000	0.653	0.347	0.347
0.106000	0.989	0.584	0.416	0.405
1.101500	0.955	0.380	0.620	0.575
2.100000	0.865	0.241	0.759	0.624
2.353000	0.843	0.208	0.792	0.635
2.364500	0.843	0.204	0.796	0.639
2.390500	0.843	0.200	0.800	0.643
2.426500	0.843	0.196	0.804	0.647
2.486500	0.831	0.196	0.804	0.636
2.539500	0.831	0.192	0.808	0.640
2.560000	0.809	0.188	0.812	0.621
3.046500	0.742	0.122	0.878	0.619
4.092000	0.438	0.033	0.967	0.406
5.108500	0.213	0.000	1.000	0.213
6.067000	0.090	0.000	1.000	0.090
8.356000	0.011	0.000	1.000	0.011
9.973000	0.000	0.000	1.000	0.000

**Table 5 jpm-11-00425-t005:** The corresponding genes for selected SNP loci.

CHR	SNP	A1	A2	MAF	Position	ENST No.	Genes	
1	rs118169127	C	T	0.05116	p13.2	ENST00000369732	OVGP1//oviductal glycoprotein 1	Downstream
2	rs112433986	-	A	0.06891	q35	ENST00000273067	MARCH4//membrane associated ring finger 4	Intron
5	rs848	A	C	0.3199	q31.1	ENST00000304506	IL13//interleukin 13	UTR-3
6	rs79454125	C	T	0.06208	q14.1	ENST00000306270	IBTK//inhibitor of Bruton agammaglobulinemia tyrosine kinase	Upstream
6	rs9490776	A	G	0.08999	q22.31	ENST00000334268	TRDN//triadin	Intron
7	rs76367287	C	A	0.07755	q36.3	ENST00000444158	uncharacterized LOC101927914	
9	rs141057871	A	G	0.1576	p22.3	ENST00000422223	FREM1//FRAS1 related extracellular matrix 1	Upstream
9	rs7847271	A	G	0.09111	q33.1	ENST00000341037	TNC//tenascin C	Intron
10	rs11255400	A	G	0.05518	p14	ENST00000344293	TAF3//TATA box binding protein associated factor 3	Intron
11	rs12797844	C	T	0.03587	q13.3	ENST00000253925	PPFIA1//protein tyrosine phosphatase, receptor type, f polypeptide (PTPRF), interacting protein (liprin), alpha 1	Upstream
11	rs7129229	T	A	0.1602	q14.3	ENST00000409404	FAT3//FAT atypical cadherin 3	Intron
13	rs9554264	T	C	0.2733	q12.2	ENST00000241453	FLT3//fms-related tyrosine kinase 3	Upstream
14	rs1959792	T	C	0.1858	q12	ENST00000546412	STXBP6//syntaxin binding protein 6 (amisyn)	Upstream intron
16	rs61401220	-	T	0.05045	q12.1	ENST00000563826	LOC101927334	
17	rs9897457	C	T	0.2128	p13.1	ENST00000330767	TMEM95//transmembrane protein 95	Downstream
17	rs3760265	T	C	0.1295	q24.2	ENST00000533854	CACNG5//calcium channel, voltage-dependent, gamma subunit 5	Intron
19	rs28933396	A	G	0.04921	q13.2	ENST00000355481	RYR1//ryanodine receptor 1 (skeletal)	Missense
23	rs149089400	A	G	0.0293	q27.3	ENST00000423667	SPANXN2//SPANX family, member N2	Upstream intron

MAF: Minor allele frequency.

## Data Availability

The datasets used and analyzed during the current study are available from the corresponding author on reasonable request.

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
