# Peer review of "Polygenic Panels Predicting the Susceptibility of Multiple Upper Aerodigestive Tract Cancer in Oral Cancer Patients"

_jpm, 2021, doi:10.3390/jpm11050425_

Round 1

Reviewer 1 Report

In the manuscript entitled “Polygenic panels predicting the susceptibility of multiple upper aerodigestive tract cancer in oral cancer patients” Huei-Tzu Chien et al. , the authors investigated head and neck cancer patients’ risks of developing multiple primaries cancers. They used the single nucleotide polymorphisms (SNPs) array to identify SNP loci which significantly vary in patients who developed multiple primary tumors (MPTs) in comparison to patients with single oral cavity cancer. Basing on this knowledge the authors built polygenic scoring model for MPTs in the upper aerodigestive tract. 

The work by Huei-Tzu Chien et al. is interesting in the field. The manuscript includes some new information, however, an additional analysis could add value to the study. There are several concerns that should be clarified or improved. 

  1. Does the coexistence of several different SNPs increase the risk of MTPs? How much? Combination of which SNPs is connected with the highest risk? 
  2. Is there any correlation between time of MTP occurrence and individual SNPs? Which SNP is connected with the earliest time of MTP occurrence? 
  3. The introduction should be complemented with information about SNPs in oral cavity squamous cell carcinoma (OSCC). 
  4. Line 108: indicate time window to “develop new primaries” 
  5. Line 165: “…426 (59.8%) had single primary cancer during follow-up.” How long was the follow-up? 
  6. The discussion is sparse in references; e.g. lack of references in lines 214, 215, 243, 247 etc. 
  7. The figures legends are insufficient; e.g. Fig.1: what the size of the dots indicates? 
  8. Lines 81-83 should be removed 

Author Response

The reply and correction were listed in the appended file. Thanks for your valuable comments!

Reviewer 2 Report

Overall, the study data are valuable. It is good designed and well written. Some minor changes are indicated below; and the major suggestion is to include female samples in the study as well. It would be more convenient. It seems that the references are not adequately placed, at least in the case of ref. 1. That should be checked. In addition, my suggestion is to enlarge the number of references.

Introduction:

Page 1, line 44: In the sentence “In Taiwan, due to the habitual use of cigarette, alcohol and areca-quid (AQ), oral 43 cavity cancer was the 4th common cancer in male.” should be indicated when was the 4th common cancer… or “was” replace with “is”. Page 1, lines 43-45: lack the references. Page 1, line 46: It seems that reference 1 doesn’t match the text. Page 2, lines 65-76: this part is too detailed and inappropriate in the introduction part. This part of the introduction should be deleted: “2. Results 80 This section may be divided by subheadings. It should provide a concise and precise 81 description of the experimental results, their interpretation, as well as the experimental 82 conclusions that can be drawn.”.

Materials and methods:

It would be better to place this sentence at the end of the paragraph Study Populaton: “This study was approved by the Institutional Review Board of Chang Gung Memo- 86 rial Hospital and undertook according to the ethical guidelines of human investigation.” Page 2, line 96.: replace “we” with “who”. 

Results:

It is not clear why to exclude female patients from the analysis?! Although they encompass very low number in the population (and why is that so?), in my opinion they should be included, along with normal female samples, which could be in larger numbers. In the case of only men patients, it should be more pointed out that the study encompasses only men; perhaps even in the title.

Discussion:

Discussion should be enlarged with more literature data and findings.

Author Response

We corrected our manuscript and replied to the reviewer's comments in the appended file. Thanks for your valuable comments.

Round 2

Reviewer 1 Report

1.Figure 2. could be smaller

2. Carefully edit the text